# Few Percent Efficient Polarization-Sensitive Conversion in Nonlinear Plasmonic Interactions Inside Oligomeric Gold Structures

**DOI:** 10.3390/s21010059

**Published:** 2020-12-24

**Authors:** Nikolay Busleev, Sergey Kudryashov, Irina Saraeva, Pavel Danilov, Andrey Rudenko, Dmitry Zayarny, Stefan A. Maier, Pham Hong Minh, Andrey Ionin

**Affiliations:** 1Quantum Electronics Division, Lebedev Physical Institute, Leninskiy Prospect 53, 119991 Moscow, Russia; busleevni@lebedev.ru (N.B.); heddwch@mail.ru (I.S.); pavel-danilov2009@yandex.ru (P.D.); aa_rudenko@mail.ru (A.R.); zayarniyda@lebedev.ru (D.Z.); ioninaa@lebedev.ru (A.I.); 2Chair in Hybrid Nanosystems, Faculty of Physics and Center for Nanoscience, Ludwig-Maximilians University of Munich, 80539 Munich, Germany; stefan.maier@physik.uni-muenchen.de; 3Department of Physics, Imperial College London, London SW7 2AZ, UK; 4Institute of Physics, Vietnam Academy of Science and Technology, 18 Hoang Quoc Viet Street, Cau Giay District, Hanoi 10000, Vietnam; phminh@iop.vast.ac.vn

**Keywords:** plasmonic sensors, resonance structures, gold nanorods, Laguerre–Gaussian beams, four-wave mixing

## Abstract

The backscattering spectra of a 500 nm thick gold film, which was excited near the 525 nm transverse localized plasmon resonance of its constituent, self-organized, vertically-aligned nanorods by normally incident 515 nm, 300 fs laser pulses with linear, radial, azimuthal and circular polarizations, revealed a few-percent conversion into Stokes and anti-Stokes side-band peaks. The investigation of these spectral features based on the nanoscale characterization of the oligomeric structure and numerical simulations of its backscattering response indicated nonlinear Fano-like plasmonic interactions, particularly the partially degenerate four-wave mixing comprised by the visible-range transverse plasmon resonance of the individual nanorods and an IR-range collective mode of the oligomeric structure. Such oligomeric structures in plasmonic films may greatly enhance inner nonlinear electromagnetic interactions and inner near-IR hotspots, paving the way for their engineered IR tunability for broad applications in chemosensing and biosensing.

## 1. Introduction

Assemblies of regularly arranged noble metal nanoelements, called plasmonic oligomers, are of interest for nonlinear plasmonics because of the broad spectra of optical-range eigenmodes and corresponding predictable spatial sets of plasmonic hotspots supported by their symmetry [1]. Such oligomers of variable spatial symmetry are suitable for tailoring nonlinear effects such as second-harmonic generation (SHG) [2,3], two-photon luminescence [4] and four-wave mixing (FWM) [5,6].

One of the possible methods for the inner structural characterization of plasmonic films is macroscopic optical spectroscopy, which can reveal their features in the form of recurring micro and nanostructures by corresponding Fano-type features in the spectra. The advantage of this kind of method is the large analyzed surface area compared with other methods such as scanning electron microscopy. In the backscattering spectra of thick gold film, multiple peaks were found, which may indicate the manifestation of different nonlinear effects.

One of the important cases of FWM [7,8] is partially degenerate four-wave mixing (PDFWM) (Figure 1). Similar to stimulated Raman scattering, low and high-frequency side-bands are called Stokes and anti-Stokes components, respectively.

In plasmonic nanostructures, the local electromagnetic field enhancement of surface plasmon polaritons (SPPs) and localized surface plasmons (LSPs) increases the efficiency of nonlinear interactions [9,10,11]. Such field enhancement can be especially strong in narrow gaps between plasmonic nanoparticles [12]. Other parameters, such as the shape and arrangement of nanoparticles, should also be taken into account [13,14].

In biosensing, localized surface plasmon resonances (LSPRs) are harnessed through various surface-enhanced processes and through resonance shifts (in the extinction spectra) induced by nearby molecules [15,16,17]. One of the attractive sensing platforms is a plasmonic array sensor, which combines the advantages of colloidal and planar substrates, including a high density of hotspots and good repeatability [18,19,20,21]. In chemical sensing, plasmonic arrays are also used for label-free refractive index gas sensing [22,23,24]. In plasmon-enhanced fluorescence (PEF) sensors, near-infrared (NIR) dyes in the “water window” (700–900 nm) are of particular interest to biosensing and bioimaging, since this spectral window allows deep penetration into biological fluids, cells and tissues. PEF is directly related to the strength of the field E generated in the vicinity of metallic surfaces. Therefore, the proper, optimal design of plasmonic sensing nanostructures, providing the maximum field intensity enhancement upon the excitation of surface plasmons, is of key importance [25,26].

In this work, we report that the self-organized oligomeric structure of nanorods in plasmonic gold films, which were considered as the simplest and most broadly available chemo and biosensing elements for decades, exhibits nonlinear Fano-type plasmonic interactions. Visually, this results in the appearance in backscattering spectra of intense sub-eV shifted sub-bands caused by PDFWM. The intensity of these sub-bands is strongly dependent on the pump laser polarization (linear, circular, radial or azimuthal). Meanwhile, near-IR hotspots could be present inside the plasmonic structure, which are highly beneficial for near-IR sensing, paving the way for their engineered IR tunability for broad applications in chemosensing and biosensing.

## 2. Materials and Methods

### 2.1. Film Deposition/Characterization and Its Backscattering Spectral Measurements

The experimental setup (Figure 2a) was similar to that in [27]. A Yb^+^-doped fiber laser Satsuma (Amplitude Systèmes, Pessac, France) with a second harmonic wavelength of 515 nm was used. Laser pulses (with a full width at a half-maximum of 300 fs) with a 10 nJ low pulse energy and repetition rate f of 10 kHz were guided through the back entrance of an upper illumination channel of a microscope–spectrophotometer MSFU-K (LOMO, Saint Petersburg, Russia, spectral range: 350–900 nm, resolution: 0.2 nm, slit width: 0.3 mm) through a 50% beam splitter (LOMO, Saint Petersburg, Russia). Then, the laser beam was focused onto the gold film in air through the microscope objective (LOMO, Saint Petersburg, Russia) with a numerical aperture (NA) of 0.65 (magnification 40×). The initial Gaussian beam with linear polarization was transformed into a donut-shaped (Laguerre–Gaussian) beam with radial or azimuthal polarization by using a commercial S-waveplate converter (Altechna, Vilnius, Lithuania). Circular polarization was achieved using a quarter-wave plate.

A gold film with a thickness of ≈500 nm was deposited on a silica glass substrate by magnetron sputtering (SC7620 Mini Sputter Coater, Quorum Technologies, Laughton, East Sussex, UK) in an argon atmosphere. The nanoscale characterization of the surface was performed by the scanning electron microscope JSM 7001F (JEOL, Tokyo, Japan) at the accelerating voltage of 10 keV. The SEM side-view image of the gold film is presented on Figure 2b. The inner structure of the film can be represented as a large number of closely spaced nanorods with a height of ≈450 nm and diameter of ≈75 nm. The SEM top-view image of the gold film was also analyzed by fast Fourier transform (FFT) (Figure 3). The FFT spatial spectrum showed a pronounced ring, which indicated an equidistant regular arrangement of the nanorods in the film [28]. The ring radius in the FFT spectrum was inversely proportional to this distance, with the ring radius corresponding to an average distance between the nanorods of ≈100 nm.

### 2.2. Numerical Simulations

The interaction of laser radiation with the nanorod trimer was numerically calculated using the finite-difference time-domain (FDTD) method. The spatial distributions of the electromagnetic field inside the gold nanorod trimer were calculated at different wavelengths for plane electromagnetic waves with linear or circular polarizations and Laguerre–Gaussian beams with radial or azimuthal polarizations. Laser radiation propagated parallel to the long axis of the nanorods. In the cases of Laguerre–Gaussian beams, the trimeric nanorods were located in the ring-like area with maximal electric field amplitude and not in the beam center (Figure 4). The height and diameter of the nanorods were set to 450 and 75 nm, respectively. The distance between the nanorods centers was set to 80 nm, and the surrounding medium was air. A perfectly matched layer (PML) boundary condition was applied to the computational domain. The dimensions of the domain were set to 2 × 2 × 1.5 μm.

## 3. Results and Discussion

### 3.1. Backscattering Spectra

Experimental polarization-sensitive backscattering spectra of the gold film are presented in Figure 5. In addition to the central peak located at the 525 nm wavelength and shifted from the 515 nm pump laser wavelength toward the transverse LSPR in the nanorods of the film, two smaller side-band Stokes and anti-Stokes peaks at 660 and 420 nm, respectively, can be observed. These wavelengths correspond to photon energies of 1.88 and 2.95 eV, which evenly differ from the laser wavelength of 515 nm (2.41 eV) by 0.53–0.54 eV, indicating a PDFWM process which produces these Stokes and anti-Stokes components.

One can assume that the incident 515 nm laser radiation is coupled to the nanorods via its LSPR, occurring at 525 nm (Figure 5). The electrical field coupled to the individual nanorods could also excite collective modes in their symmetrical oligomers present in the inner structure of the gold film. The experimental indication of such hybridization is the Fano-like waveform of the side-band spectral components in Figure 5; specifically, the dip located at the wavelength of 650 nm. Such an asymmetrical dip could indicate the Fano resonance, which occurs in oligomer systems as a result of the destructive interference of the localized dark mode and the collective broad bright mode [29]. These plasmonic modes have wavelengths of 2350–2300 nm (0.53–0.54 eV). Moreover, comparing the backscattering spectra for each polarization in Figure 5, side peaks are more pronounced in the cases of the radial and azimuthal polarizations. These polarization types are more efficient when coupling to individual nanorods (radial polarization, high-NA appearance of a longitudinal electric field component [30,31,32]) and to their symmetrical oligomers (both radial and azimuthal polarizations). Such differences in the spectra, depending on the laser polarization type, can be explained in terms of the different collective plasmonic modes that arise in the inner structure of the gold film (Figure 6).

In the quantitative aspect, the central peak value of the backscattering intensity (being close to saturation) in Figure 5 equals *I*_exc_ ≈ 20 × 10^3^ arb. units (not shown on Figure 5); the magnitude of the side-band peaks in cases of azimuthal and radial polarization equals *I*_PDFWM_ ≈ 300–400 arb. units. Thus, the PDFWM efficiency can be evaluated as follows:(1)$$\eta =I_{PDFWM}/I_{exc}$$
In this case, its value approaches to 10^−2^ (~1%). For comparison, the FWM efficiency in other nanoscale systems reaches ~10^−5^ for gold nanotip [33,34] and germanium nanodisk [35] and 10^−4^ for a metasurface made of silver nanoantennas with embedded diamond nanoparticles [36].

### 3.2. Numerical Simulations

Since the FFT spectrum (Figure 3b) shows the isotropic ordering (close packing) of gold nanorods in the film with hexagonal symmetry, a trimer appears as the elementary structural block. The normalized electric field amplitude was calculated in the entire computational domain. Maximal values of the squared electric field amplitude ${\left|E\right|}^{2}$ calculated inside the nanorod trimer and the experimental backscattering intensity of the Stokes component are presented as spectra for different wavelength values in Figure 7. The calculated ${\left|E\right|}^{2}$ magnitude demonstrates a spectral peak located in the range of 2000–2200 nm (0.56–0.62 eV) and represents the collective plasmonic mode in the trimer responsible for Stokes and anti-Stokes components (side-band peaks in Figure 5). The corresponding spatial distributions of the normalized electric field amplitude are presented in Figure 8. As can be seen from this figure, the maximum of the amplitude is located between nanorods.

Our interpretation of the experimental results is based on the close analogy to the well-known phenomenon of the nonlinear optical FWM generation of side-bands in a nonlinear medium [37]. In our case, the normally incident laser radiation is resonantly and efficiently converted into localized transverse plasmons of each illuminated, vertically aligned gold nanorod. The resulting strong, localized electric field excites collective modes of the nanorod ensemble, and this coupling creates the observed side-bands in our spectra. The NIR plasmonic mode, revealed in the numerical simulations, apparently defines the frequency shift of each side-band in PDFWM (Figure 1). This nonlinear interaction occurs via the destructive interference of the localized dark mode (transverse LSPR of nanorods) and the collective broad bright mode (NIR). As SEM images are the only source of information of the nanorods’ geometrical parameters, their dimensions were evaluated approximately rather than very exactly. This leads to an imperfect match between the experimental and numerical simulations results.

## 4. Conclusions

In this work, we experimentally observed intense visible-range Fano-type side-band features in backscattering spectra of the 500-nm gold film, presented by the self-organized oligomeric structure of nanorods, under its 515 nm 300fs laser excitation with different laser polarizations: linear, circular, radial and azimuthal. Such a partially-degenerate four-wave mixing mechanism was related to laser coupling to the transverse localized plasmon resonance in the nanorods, hybridized in the near-field with the near-IR collective mode of the nanorod trimer, as the elementary structural block. The corresponding conversion efficiency was evaluated as high as 10^−2^. Even though the visual appearance of the effect was in the visible range, due to the symmetry of the involved oligomers, there were three hotspots of the broad, bright collective mode, which are promising for near-IR sensing, while also paving the way for their broad IR tunability, which can be engineered via unit cell scaling and packing for applications in chemosensing and biosensing.

## Figures and Tables

**Figure 1 sensors-21-00059-f001:**
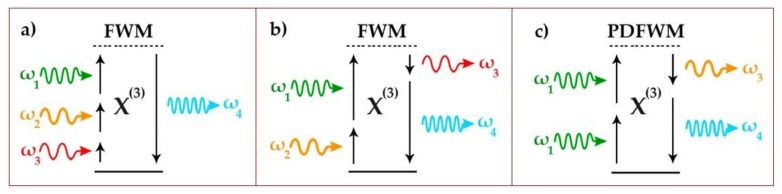
Energy level schemes of (**a**,**b**) four-wave mixing (FWM) and (**c**) partially degenerate four-wave mixing (PDFWM).

**Figure 2 sensors-21-00059-f002:**
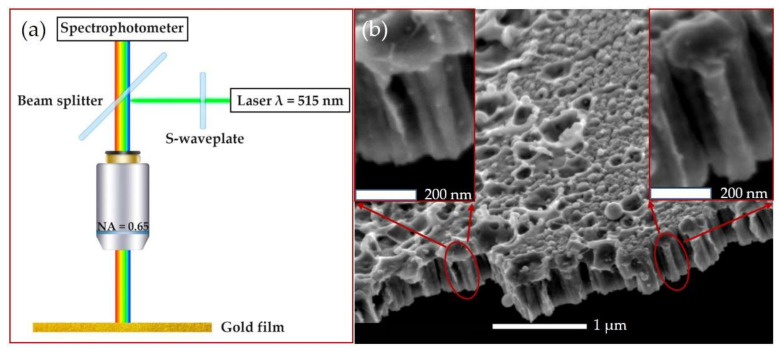
(**a**) The schematic of the experimental setup. (**b**) The SEM side-view image of the gold film inner structure at different magnifications. The nanorod-like element is shown in the red circle.

**Figure 3 sensors-21-00059-f003:**
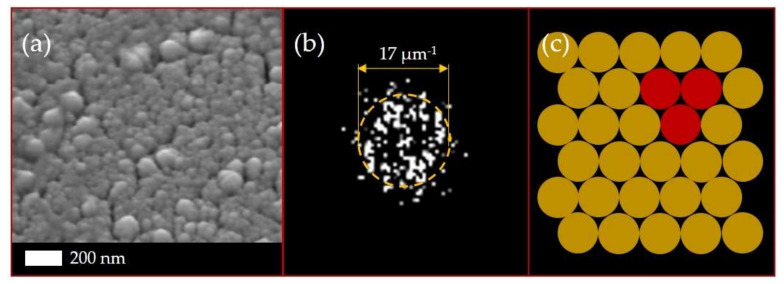
(**a**) The SEM top-view image of the gold film and (**b**) its fast Fourier transform (FFT) spectrum. The ring-pattern radius corresponds to the average distances between the structure elements (nanorods). (**c**) Schematic illustration of oligomeric structure. The building block in the form of a trimer is highlighted.

**Figure 4 sensors-21-00059-f004:**
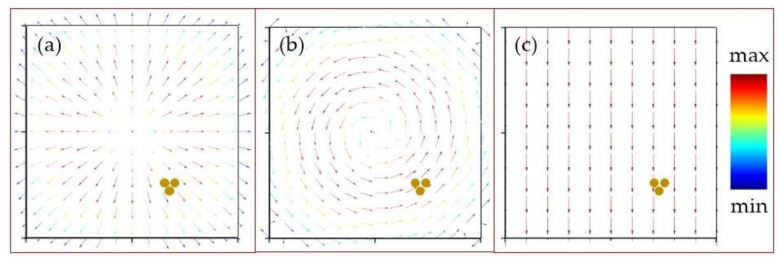
Vector plot of electric field amplitude in the cases of (**a**) a Laguerre–Gaussian beam with radial polarization, (**b**) a Laguerre–Gaussian beam with azimuthal polarization and (**c**) a plane wave with linear polarization. The location of the nanorod trimer in each case is shown by the three yellow dots.

**Figure 5 sensors-21-00059-f005:**
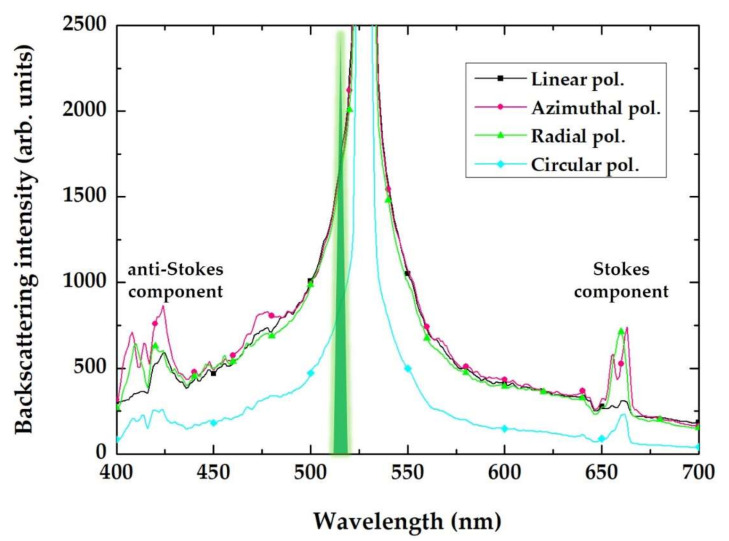
Experimental backscattering spectra of the bare gold film for the 515 nm, 300 fs laser pulses with various polarizations (as indicated in the frame), showing the maximum scattering at the plasmon resonance wavelength and additional Stokes/anti-Stokes side-band components.

**Figure 6 sensors-21-00059-f006:**
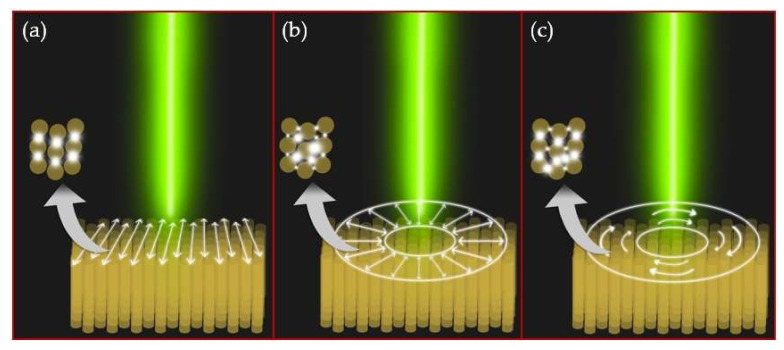
Schematic differences in collective plasmonic modes excited in the oligomeric structure of nanorods, depending on the polarization type: (**a**) linear, (**b**) radial and (**c**) azimuthal. The electric near-field is localized in narrow gaps between gold nanorods. The arrangement of hotspots depends on the spatial location of the oligomer in relation to the center of the laser beam in the case of radial or azimuthal polarization.

**Figure 7 sensors-21-00059-f007:**
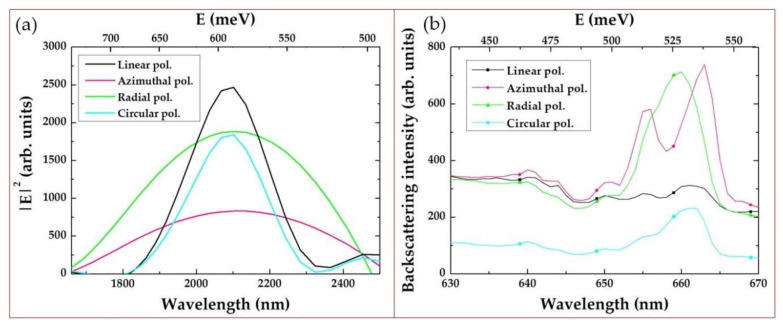
(**a**) Maximal values ${\left|E\right|}^{2}$ calculated inside the nanorod trimer for various polarizations depending on the wavelength (bottom *x*-axis) and photon energy (top *x*-axis). (**b**) Experimental backscattering spectra of the Stokes-component intensity. Top *x*-axis shows difference in photon energy with laser pump (515 nm).

**Figure 8 sensors-21-00059-f008:**
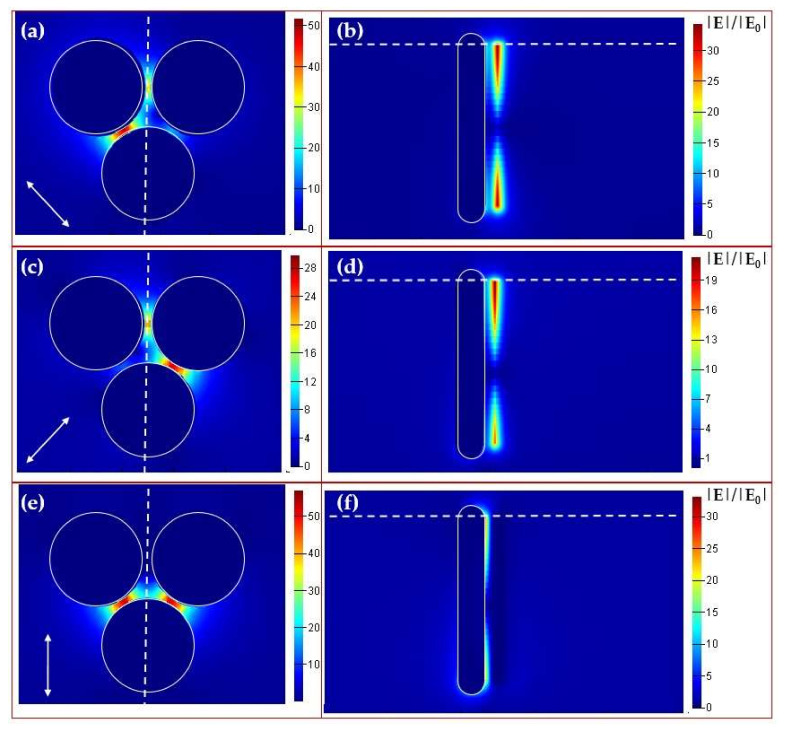
Spatial distributions of the electric field amplitude for radial (**a**,**b**), azimuthal (**c**,**d**) and linear (**e**,**f**) polarization and λ = 2100 nm inside the nanorod trimer. (**a**,**c**,**e**) Top view in the cross-section plane shown by the white dashed line in (**b**,**d**,**f**). The white arrow shows the local polarization direction. (**b**,**d**,**f**) Side view in the cross-section plane shown by the white dashed line in (**a**,**c**,**e**). Electric field localization between nanorods can be observed. The color scale is given for the normalized magnitude |E|/|E_0_|. The electric field distribution in (**a**) is asymmetrical because of the location of the trimer in the bottom-right sector of the Laguerre–Gaussian beam with radial polarization. The trimer is not exactly symmetrical in relation to the polarization vector in this location.

## Data Availability

The data presented in this study are available on request from the corresponding author.

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
