# Peer review of "Few Percent Efficient Polarization-Sensitive Conversion in Nonlinear Plasmonic Interactions Inside Oligomeric Gold Structures"

_sensors, 2020, doi:10.3390/s21010059_

Round 1
Reviewer 1 Report
In this contribution, the backscattering spectra of a 500-nm thick gold film is investigated. The laser pulses with linear, radial, azimuthal, and circular polarizations, revealed a few-percent conversion into anti- and Stokes side-band peaks. Spectral features of its backscattering response indicated nonlinear Fano-like plasmonic interactions.
I ask the author to address the following questions before publication.
- The Introduction covers a wide area of knowledge- It would be better if the author focuses more on the specific topic of this work.
- Some sentences are too large and difficult to understand. For example, line 132-135 and line 136-140
- I recommend to re-formulate some sentences such as line 71-73 and line 106
- A couple of spectrums are presented in the manuscript with different units as x-axis [cm-1 ], [nm], and [eV]. I recommend choosing one to make it easier for a reader to follow.
- Please explain why trimer symmetry is cohoused for the simulation?
- The author in line 103 wrote, “Spatial distributions of the electromagnetic field inside the gold nanorod trimer were calculated ...”. Is it only one point inside it? Please indicate exactly where is this point and how it is chosen?
- About simulation presented in Fig7: Is it done only one trimer as shown in Fig4 or an array as shown in Fig6?
- Why there is a shift between experiment and simulation in Fig7a and Fig7b?
- Do you have a reference or theory to relate this experiment and simulation to each other?
- Fig8 presents the simulation result for radial polarization. Have you done a similar study for other polarizations (azimuthal and linear)?
- Why nanorods presented in Fig8 have no circle shape?
- Can the author explain, why the E field distribution presented in Fig8a is not symmetrical?
Reviewer 2 Report
The manuscript demonstrates experimentally measured highly efficient partially degenerate four-wave mixing in oligomers of Au nanorods. Although plasmonic thin films have been studied for decades, the Authors managed to show quite rarely observed pronounced nonlinear features, which are of definitive interest for chemo- and biosensing.
I recommend the manuscript for publication in Sensors, although suggest the Authors to address a few minor recommendations:
- The Authors refer to the plasmonic system under consideration in a variety of similar yet in overall confusing terms: "oligomers", "thin films", "nanorod array", "oligomeric structure", "island-type films" and etc. I suggest to eliminate some of them and end up with something more unified and clearer to the Reader;
- Why numerical results consider a trimer, but not something larger and quasi-randomly arranged to get closer similarity with the experiment? The justification for consideration of a trimer is missing at the beginning of Sec.3.2;
- How numerical results in Sec.3.2 explain or elaborate the experimental data? Why one can immediately infer the direct relation of the peak in Fig.7a to Stokes and anti-Stokes peaks in Fig.5? A bit thorougher presentation and explanation will help the Reader to understand;
- Cross-sections of nanorods in Fig.8 seem to be elliptical contrary to the Reader's expectations to observe circles of radius 75nm as stated in the manuscript;
- Minor grammatical issue in line 34, should be " ...are suitable...", not "...suitable..."
Round 2
Reviewer 1 Report
Thank you, authors, for their effort however still some points I recommend authors to do,
Please add Response 5, Response 8, and Response 10 to the manuscript, Sec. 3.2.
Author Response
Point 1: Thank you, authors, for their effort however still some points I recommend authors to do,
Please add Response 5, Response 8, and Response 10 to the manuscript, Sec. 3.2.
Response 1: Response 5 was added (lines 161 and 168).
Response 8 was added (Figure 8c–f).
Response 10 was added (lines 181–184).